# REVISITING MAE PRE-TRAINING FOR 3D MEDICAL IMAGE SEGMENTATION

## ABSTRACT

Self-Supervised Learning (SSL) presents an exciting opportunity to unlock the potential of vast, untapped clinical datasets, for various downstream applications that suffer from the scarcity of labeled data. While SSL has revolutionized fields like natural language processing and computer vision, their adoption in 3D medical image computing has been limited by three key pitfalls: Small pre-training dataset sizes, architectures inadequate for 3D medical image analysis, and insufficient evaluation practices. We address these issues by i) leveraging a large-scale dataset of 44k 3D brain MRI volumes and ii) using a Residual Encoder U-Net architecture within the state-of-the-art nnU-Net framework. iii) A robust development framework, incorporating 5 development and 8 testing brain MRI segmentation datasets, allowed performance-driven design decisions to optimize the simple concept of Masked Auto Encoders (MAEs) for 3D CNNs. The resulting model not only surpasses previous SSL methods but also outperforms the strong nnU-Net baseline by an average of approximately 3 Dice points. Furthermore, our model demonstrates exceptional stability, achieving the highest average rank of 2 out of 7 methods, compared to the second-best method's mean rank of 3. Our code is made available here.

## 1 INTRODUCTION

In recent years, the concept of Self-Supervised Learning (SSL) has emerged as a driving factor in data-rich domains, enabling large-scale pre-training that facilitates the learning of robust and transferrable general-purpose representations (Assran et al., 2023; Oquab et al., 2023; He et al., 2022). This paradigm shift has been instrumental in advancing various fields, particularly in domains with abundant labeled data like NLP or natural vision. In the domain of 3D medical image computing, this trend has not caught on.

Currently, the domain is either focused on training from-scratch, mainly using the nnU-Net framework by Isensee et al. (2021), or using supervised pretraining, which is limited by the cost associated with annotated data (Wasserthal et al., 2023; Ulrich et al., 2023; Huang et al., 2023). The usage of supervised pretraining implies the willingness of the domain to adopt pretraining and calls to question the currently established SSL methods in the domain. We believe this lack of widespread adoption of previously established methods can be attributed to three major pitfalls of previous SSL research in this field:

**P1 - Limited dataset size:** Many SSL approaches have been developed on very few unlabeled volumes, often being trained on fewer than 10,000 images Zhou et al. (2021); Wu et al. (2024); Tang et al. (2024; 2022); Zhuang et al. (2023); Hatamizadeh et al. (2021), almost approaching scales of supervised dataset sizes. These datasets tend to be pooled from publicly available annotated datasets, as larger datasets pose a greater hurdle to acquire. While many hospitals possess 3D medical images in the millions, they are locked away from the public due to patient privacy concerns. While some larger open-source datasets exist, e.g., the Adolescent Brain Cognitive Decline (ABCD) dataset of the NIH (N=40k) or the UK Biobank (UK-BB) (N=120k), they restrict access pending an internal review board's approval, posing a hurdle for open science. More recently, the UK-BB stopped allowing downloading data to local hardware, signaling a public decrease in the community's willingness to share data.

**P2 - Outdated Backbones:** Many studies develop their SSL method on non-state-of-the-art architectures, e.g., utilizing transformers (Wu et al., 2024; Tang et al., 2022; Wang et al., 2023; Chen et al., 2023). While transformers are prevalent in the 2D natural imaging domain (Dosovitskiy, 2020), recent architectures leveraging attention (Vaswani, 2017) have so far not been able to reach state-of-the-art performance in 3D medical segmentation. In fact, well-configured 3D U-Net (Çiçek et al., 2016; Ronneberger et al., 2015) inspired CNNs dominate 3D medical image segmentation, outperforming transformer-based models by a large margin Isensee et al. (2024). This underscores the need for SSL methods that can be seamlessly integrated with CNNs to harness their full potential in medical image analysis on downstream tasks.

**P3 - Insufficient Evaluation:** Existing methods often lack rigorous evaluation, masking the methods' efficacy (or lack thereof). This is represented through: i) Evaluating on too few datasets to show generalization of the pre-training ii) Stacking multiple contributions, e.g., novel architecture with new pre-training, which does not allow one to draw conclusions if the pre-training is effective without the architecture (Wang et al., 2023) iii) Comparing to bad from-scratch baselines, like a badly configured, outdated Çiçek et al. (2016) model instead of a well-configured 3D nnU-Net CNN baseline (Isensee et al., 2021). iv) Evaluating their method on seen data they pre-trained on. We want to emphasize that we do not intend to point fingers, but to raise awareness that evaluation matters and insufficient evaluation can lead to a lack of clarity which methods are the best. This observation is similar to the recent study by Isensee et al. (2024), that shows that this is also prevalent in the field of medical image segmentation when training from-scratch.

In this paper, we carefully avoided all these pitfalls while exploring the Masked Auto Encoder (MAE) paradigm for 3D CNNs, with the recent adaptations introduced by Tian et al. (2023); Woo et al. (2023), and highlight that MAEs are exceeding the current state-of-the-art SSL methods in 3D medical image segmentation given proper configuration. Our contributions can be summarized as follows:

1. We evade Pitfall 1 by leveraging a dataset collection of 44k 3D MRI volumes to develop our self-supervised pretraining, exceeding the majority of 3D medical image segmentation SSL methods in scale (Zhou et al., 2021; Wu et al., 2024; Tang et al., 2024; 2022; Zhuang et al., 2023; Hatamizadeh et al., 2021).

2. We evade Pitfall 2 by utilizing the state-of-the-art Residual Encoder U-Net CNN architecture from Isensee et al. (2024) as the backbone. Moreover, we use this backbone for all baseline SSL methods - allowing us to quantify the utility of the pre-training scheme for pre-training CNNs.

3. We evade Pitfall 3 by employing 5 development and 7 testing datasets, spanning a diverse set of downstream targets. This includes head and neck organs and pathologies, datasets with novel image modalities not seen during pre-training, and datasets of the same pathology acquired at different centers.

By subsequently optimizing MAEs for CNNs based on performance on the development pool, we propose the novel sparse MAE inspired pre-training paradigm, unleashing the potential of huge amounts of not annotated data in the medical domain. We show that the choice of the best fine-tuning strategy is crucial. Moreover, we evaluate our method across a range of critical 3D medical scenarios, including low-data regimes, accelerated fine-tuning schedules, and generalization across unseen centers and modalities.

## 2 DEVELOPMENT FRAMEWORK

The goal of this paper is to develop a robust SSL pretraining method. Due to the limited prior work in the 3D medical domain, many design choices need to be made. We address this by sequentially validating each methodological contribution on five downstream development datasets before testing the final configuration on eight untouched test datasets. To reduce the search space and disentangle the effects of SSL pre-training from other basic design choices, we choose to keep some parameters fixed based on best practices in the domain:

(**i**) The architecture used is always the same state-of-the-art residual encoder U-Net architecture (ResEnc U-Net) (Isensee et al., 2024). (**ii**) The input patch size is [160x160x160]. (**iii**) All images are resampled to the target spacing of [1x1x1] mm$^3$ (Roy et al., 2023). (**iv**) All images are z-score

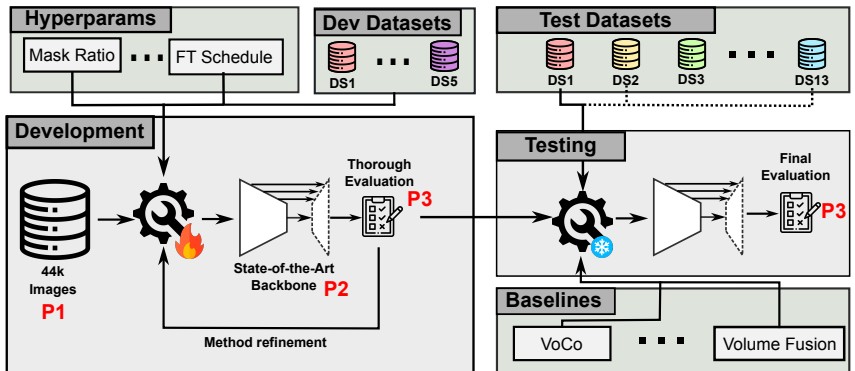

Figure 1: **Overview of the model development pipeline.** During pretraining, we carefully develop our method with 44k images (P1), with a state-of-the-art residual encoder U-Net (P2) and evaluate it on dive development datasets (P3). In the testing phase we utilize a total of thirteen datasets (P3) to verify the utility of our developed method.

normalized to zero mean and unit variance (Isensee et al., 2021). (**v**) As optimizer, we use SGD with a decreasing poly-learning rate (Chen et al., 2018) following nnU-Net (Isensee et al., 2021). (**vi**) We always employ random sampling during pre-training, irrespective of the prevalence of the different MR modalities.

**Pre-training Dataset**   To develop our pre-training method, we utilize a proprietary brain MRI dataset sourced from over 44 centers containing over 9k patients comprising a total of approx. 44k 3D MRI scans. Due to the variety of data sources, this dataset contains images from more than 10 different MR scanners, various MR modalities and a diverse patient population. For more details on the distributions we refer to Figure 2.

Since this data is sourced directly from clinical examinations, it includes empty or broken images, poor-quality images and so-called scout scans used to determine the field of view of the patient in the MR. Since these scans are not used in diagnostics, these images are filtered by discarding (**a**) images with a field of view of $< 50mm$ in any axis, (**b**) images with a spacing $> 6.5mm$ in any direction and (**c**) images of file size $< 200kb$, which indicate an empty image. Moreover due to the low quantity of MR Angiography, Susceptibility weighted Images (SWI) and Proton Density (PD) weighted images, we restrict our training data to only include T1, T2, T1 FLAIR and T2 FLAIR images, resulting in our final pre-training dataset of $39,168$ MR images.

**Development Datasets**   After pre-training we fine-tune on five datasets and calculate the average Dice Similarity Coefficient (DSC) per dataset to evaluate the effectiveness of the pre-training. The multiple datasets are essential to ensure that our design choices do not overfit to a specific MRI modality or pathological target. Specifically, we utilize:

1. **MS FLAIR (D1)**: Consensus delineations of multiple sclerosis (MS) lesions on T2-weighted FLAIR images(Muslim et al., 2022).

2. **Brain Mets (D2)**: Brain metastases imaged through T1, contrast enhanced gradient echo T1ce, contrast enhanced spin echo T1 and a T2 FLAIR sequence acquired in the Stanford University Hospital (Grøvik et al., 2020).

3. **Hippocampus (D3)**: The Hippocampus dataset, task 4 of the medical segmentation decathlon (MSD) (Antonelli et al., 2022), contains delineations of the anterior and posterior Hippocampus in T1 weighted MRI (Simpson et al., 2019).

4. **Atlas22 (D4)**: Anatomical Tracings of Lesions After Stroke (ATLAS) on T1 weighted images. We use the Atlas R2.0 dataset from Liew et al. (2022).

5. **CrossModa (D5)**: Delineations of intra-meatal and extra-meatal vestibular schwannoma tumors and cochlea delineations in contrast enhanced T1 weighted MRI (Dorent et al., 2023).

Of all these datasets, we set aside a hold-out test set comprising 20% of all images before start of the method development process. The remaining images were further split 80/20 into training and validation sets for the development process.

**Test Datasets**  Additionally, eight hold-out test sets were used to evaluate the efficacy of our learned representations when fine-tuning them to segment other target structures.

1. **Cosmos (D6):** This dataset features carotid vessel wall segmentation and atherosclerosis diagnosis, of which we use the contours to evaluate segmentation performance (Chen et al., 2022).

2. **HaNSeg (D7):** This dataset contains segmentations of 30 organs at risk (OAR), with associated T1 MRI and CT, of which we only use the MR images for model development (Podobnik et al., 2023).

3. **Isles22 (D8):** This dataset contains annotations of ischemic stroke lesions, with associated diffusion-weighted imaging (DWI), apparent diffusion coefficient (ADC) and T2 FLAIR (Hernandez Petzsche et al., 2022).

4. **HNTS-MRG24(D9):** T2-weighted MR images of pre-treatment oropharyngeal cancer and metastatic lymph nodes with associated annotations (Wahid et al., 2024).

5. **BraTS Africa (D10):** This dataset contains 1.5 Tesla T1, T1ce, T2, and T2 FLAIR MRIs of glioblastoma and high-grade gliomas imaged in Nigeria.

6. **T2 Aneurysms (D11):** This proprietary dataset contains 240 T2 MRI images of segmented brain aneurysms with the surrounding brain tissue.

7. **TOF Angiography Aneurysms (D12):** This proprietary dataset contains 144 time-of-flight MR-angiography images of segmented brain aneurysms with the surrounding brain tissue.

8. **BraTS Mets (D13):** This dataset holds brain metastases segmentations on T1ce MRIs, similar to **D2**. However, instead of fine-tuning on it we use it to measure generalization when inferring models trained on **D2** on it.

For all test datasets that are fine-tuned on (**D6-D11**) we use an 80/20 split for fine-tuning and for testing, as we fine-tune each method only once without any interventions. For **D13** all data is used, as no training is conducted.

## 3 REVISITING 3D MAEs

Masked autoencoders (MAEs) are a well-established pre-training paradigm in the natural imaging domain and in the medical image segmentation domain for transformers. In this section, we investigate this paradigm and optimize it for 3D medical image segmentation using a ResEnc U-Net architecture of Isensee et al. (2024).

**Default parameters**  MAEs are trained by masking an input image to a certain degree and training the network to reconstruct the occluded regions, minimizing deviations betweem reconstruction and original image. In our experiments, we train the MAE with a L2-Loss in the z-score normalized voxel space and only calculate the reconstruction loss where regions were masked. Moreover, we do not remove skip-connections, following the general consensus of Woo et al. (2023), Tian et al. (2023), and He et al. (2022). The default hyperparameters (as used by the model denoted in gray in Table 1) are learning rate 1e-2, weight decay 3e-5, batch size 6, SGD optimizer with Nesterov momentum 0.99, masking ratio of 75% trained with a PolyLR schedule for 250k steps (this represent 1000 epochs in the nnU-Net framework) and minor spatial augmentations of affine scaling, rotation and mirroring.

**Sparsification**  When masking the input image, CNNs are not able to ignore the masked regions in the same manner as transformers can. To address this, Tian et al. (2023) proposed to adapt the CNN architectures to better fit the sparse inputs: (**a**) **Sparse Convolutions and Normalization:** Through the receptive field of convolutions masked-out regions are iteratively eroded from their

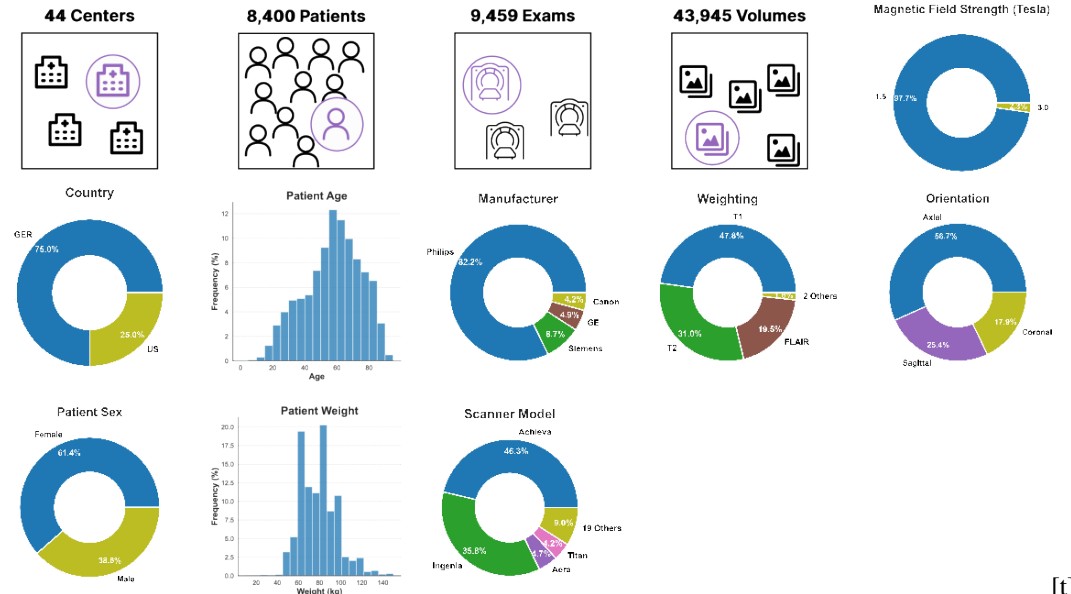

[t]

Figure 2: Hierarchical composition of the nnSSL foundation model's pre-training dataset. (a) Geographic distribution of 44 participating centers. (b) Patient demographics (age, sex, weight). (c) Scanner specifications (magnetic field strength, manufacturer, scanner) of examinations. (d) Acquisition parameters (orientation, contrast administration, weighting) of image volumes.

boundaries. By re-applying the masked regions after every convolution this problem can be resolved. Moreover, the masks can introduce a problematic shift in the normalization layer statistics due to the introduced zero values. To resolve this normalization is constrained to only consider the non-masked values. (**b**) **Mask Token:** Instead of feeding the light-weight decoder feature maps with zeroed mask regions, the regions are densified by filling them with a learnable mask token, simplifying the reconstruction task of the decoder. (**c**) **Densification Convolution:** After filling the masks with the Mask Token and before passing the feature maps to the decoder, a [3x3x3] convolution is applied to the feature maps at every resolution except the highest resolution to prepare the representations for decoding.

Results of these changes are visualized in Table 1a. The adaptations are introduced iteratively, meaning the 'MaskToken' ablation is only applied together with the Sparse Convolutions and Normalization. It can be observed that the full set of adaptations improves performance by an average of 0.3 DSC points across our development datasets. Subsequently, all adaptations are kept and the following evaluations are presented with these changes applied.

**Masking strategy** The masked region is determined by sampling randomly in the CNN's bottleneck of shape [5x5x5] and up-sampling these regions to the input resolution, to ensure the masks align in the bottleneck of the CNN architectures. This results in masking regions of [32x32x32] voxels of non-overlapping regions in the input. As a sampling strategy, we follow random masking, as previous work showed no benefit of structured masking for images nor videos (He et al., 2022; Feichtenhofer et al., 2022). In the scope of the development phase we explore 5 static masking ratios between 30 and 90%, and evaluate a dynamic masking ratio randomly masking between 60% and 90%.

Results are presented in Table Table 1b and highlight that the masking ratios of 60%, 75%, and the dynamic masking ratio of 60% to 90% perform equally well. Due to the highly similar performance, we choose to proceed with a dynamic masking ratio over the static masking ratio, due to expecting this masking to be more difficult to learn and with the upcoming scaling experiment in mind. We refer to this model as Spark3D-Base (**S3D-B**= going forward.

| Configuration | D1 | D2 | D3 | D4 | D5 | Avg. D1-D5 |
|---|---|---|---|---|---|---|
| Base | 49.96 | 72.62 | **89.03** | **63.45** | 81.67 | 71.35 |
| + Sparse Conv. + BN. | 50.34 | 73.34 | 88.91 | 62.96 | 81.25 | 71.36 |
| + w/ Mask Token | 50.02 | 73.21 | 88.92 | 62.84 | **81.83** | 71.37 |
| + w/ Dens. Conv. | **51.02** | **74.07** | 88.91 | 62.81 | 81.50 | **71.66** |

(a) **Sparsification:** Introducing all sparsification adaptations showed best results, with most gains from adding the additional densification conv layer between encoder and decoder.

| Mask ratio | D1 | D2 | D3 | D4 | D5 | Avg. D1-D5 |
|---|---|---|---|---|---|---|
| 30 % | 50.25 | 71.37 | 88.90 | 63.18 | 81.70 | 71.08 |
| 45 % | 50.60 | 70.83 | 88.97 | 63.27 | **81.73** | 71.08 |
| 60 % | 50.62 | 73.56 | 88.96 | **63.37** | 81.48 | 71.60 |
| 75 % | 51.02 | **74.07** | 88.91 | 62.81 | 81.50 | **71.66** |
| 90 % | 50.56 | 72.51 | **88.98** | 62.41 | 81.49 | 71.19 |
| U[60%-90%] | **51.49** | 74.01 | 88.83 | 62.39 | 81.54 | 71.65 |

(b) **Masking ratio:** Masking ratios between 60% and 75% worked best when choosing static ratios with a dynamic range including higher masking ratios performing the best.

| Pretraining | D1 | D2 | D3 | D4 | D5 | Avg. D1-D5 |
|---|---|---|---|---|---|---|
| No Dyn. | 45.56 | 72.26 | 88.80 | 60.44 | **82.61** | 69.93 |
| No Fixed | 49.37 | 69.13 | 88.78 | 60.74 | 81.33 | 69.87 |
| VoCo | 50.35 | 67.20 | 88.22 | 57.82 | 80.29 | 68.77 |
| VF | 49.93 | 69.58 | 88.83 | 61.75 | 81.48 | 70.31 |
| MG | 50.50 | 71.14 | 88.83 | 63.29 | 82.15 | 71.18 |
| **S3D-B (ours)** | **51.49** | **74.01** | 88.83 | 62.39 | 81.54 | 71.65 |
| **S3D-L (ours)** | 51.42 | 72.84 | **89.09** | **63.30** | 82.15 | **71.76** |

(c) **Compound scaling:** When comparing our scaled S3D-L, trained for more iterations with higher batch size and lr, only a modest increase in performance can be observed over the much more compute-efficient S3D-B. Moreover, when comparing models on the same scale S3D-B exceeds all baselines on our development validation datasets.

Table 1: **Development Experiments:** During development, we evaluate the Average DSC on all development datasets to quantify the best configuration. Methods of the same configuration are denoted through common colors.

**Scaling**    MAEs are known to benefit from scaling. We evaluate the effect of compound scaling by increasing the batch size by x8 to $48$, the learning rate to $3e-2$, and the iterations by x4 to $1M$. We refer to this model as S3D-Large (**S3D-L**) to denote the higher compute resources, but note that the architecture remains identical, to maintain easy adaptation of the parameters. Results are presented in Table 1c. It can be observed that this x32 increase in compute only resulted in a slight increase in performance of 0.1 DSC points.

**Fine-tuning strategy**    Given a pre-trained model, a crucial question arises: Which weights to transfer and how to schedule the fine-tuning? We investigated various different schedules. Regarding weight transfer we investigated transferring (i) both the encoder and decoder ▶◀, or (ii) only the encoder ▶◀ with a randomly initialized decoder. Regarding the fine-tuning schedule, we investigate whether to use a learning rate warm-up of 12.5k steps, ramping the learning rate up to the maximum LR. When only transferring the encoder, an additional warm-up of only the decoder is investigated to adapt the randomly initialized decoder to the pre-trained encoder. In some configurations, this results in two learning rate warm-ups of 12.5k steps each. Additionally, we investigate whether to keep the encoder frozen ▶◀ for the entire fine-tuning process, or to fine-tuning encoder weights as well. Lastly, we investigate whether to decrease the peak learning rate to $1e-3$, $1e-4$ or to keep it at the default of $1e-2$.

| Transfer | 1. Warm-Up | 2. Warm-Up | Tr. Stage | Max. LR | D1 | D2 | D3 | D4 | D5 | Avg |
|---|---|---|---|---|---|---|---|---|---|---|
| 🔴 | - | - | 🟢 | 1e-2 | 45.56 | 72.26 | 88.80 | 60.44 | 82.61 | 69.93‡ |
| 🔴 | - | - | 🟢 | 1e-2 | 49.37 | 69.13 | 88.78 | 60.74 | 81.33 | 69.87 |
| 🟢 | - | - | 🟢 | 1e-2 | 50.37 | 70.64 | 88.61 | 61.51 | 81.91 | 70.61 |
| 🟢 | - | - | 🟢 | 1e-3 | 49.98 | 71.04 | 88.68 | 61.45 | 82.12 | 70.65 |
| 🟢 | - | 🟢 | 🟢 | 1e-2 | 49.84 | 72.56 | 88.45 | 62.16 | 81.75 | 70.95 |
| 🟢 | - | 🟢 | 🟢 | 1e-3 | 51.54 | 72.74 | 88.85 | 62.44 | 82.33 | _71.58_ |
| 🟢 | - | 🟢 | 🟢 | 1e-4 | 50.66 | 72.98 | 88.68 | 62.73 | 82.09 | 71.43 |
| 🟢 | - | - | 🟢 | 1e-2 | 50.81 | 68.55 | 88.52 | 60.73 | 80.77 | 69.87 |
| 🟢 | - | - | 🟢 | 1e-3 | 50.11 | 73.90 | 88.58 | 61.82 | 81.48 | 71.18 |
| 🔴 | - | 🔴 | 🔴 | 1e-2 | 51.39 | 72.06 | 88.65 | 61.70 | 81.78 | 71.12 |
| 🔴 | - | 🔴 | 🔴 | 1e-3 | 51.04 | 72.72 | 88.91 | 62.84 | 81.19 | 71.34 |
| 🟢 | 🔴 | 🟢 | 🟢 | 1e-2 | 48.81 | 72.92 | 88.84 | 62.46 | 82.04 | 71.02 |
| 🟢 | 🟢 | 🟢 | 🟢 | 1e-3 | 51.42 | 72.84 | 89.09 | 63.30 | 82.15 | **71.76** |
| 🟢 | 🔴 | 🟢 | 🟢 | 1e-4 | 50.13 | 72.26 | 88.72 | 61.70 | 81.93 | 70.95 |

Table 2: **Fine-tuning matters.** We compare various combinations of weight transfer and fine-tuning schedules. Transfer: 🟢 Transfer all, 🔴 Transfer encoder only. *Warm-Up and Training:* 🔴 Only decoder weights trained, 🟢 Encoder and decoder weights trained. ‡: nnU-Net default (Dynamic)

Results are presented in Table 2 and allow three important observations to be made: (i) **Warm-up stages are essential:** Not applying a warm-up step significantly reduces performance. Including a warm-up for both the encoder and decoder boosts accuracy by 0.6 to 1 DSC points. (ii) **Learning rate adjustments matter:** Reducing the peak learning rate to 1e-3 during fine-tuning consistently yields better results than the default 1e-2, with the best performance seen when fine-tuning both the encoder and decoder with lower learning rates. (iii) **Freezing encoder weights is detrimental:** The encoder should not remain fixed during fine-tuning. Allowing the encoder to be fine-tuned improves the model's performance compared to when only the decoder is fine-tuned.

## 4 RESULTS AND DISCUSSION

We compare our final models **S3D-B** and **S3D-L** against **VoCo** (Wu et al., 2024), VolumeFusion **(VF)** by Wang et al. (2023), as well as **MG** Models Genesis (Zhou et al., 2021). The baselines are pre-trained using the same framework on the same data with the same backbone and the same hyperparameters - where possible - and are scaled to fully utilize an A100 40GB GPU, optimized for 250k steps. We provide explicit baseline method and configuration details in Appendix A. Moreover, we compare against two from-scratch baselines. The first, 'No (Dyn.)', represents a non-pretrained (i.e., from scratch) default nnU-Net `3D fullres` architecture that was planned and preprocessed on each downstream dataset individually, potentially resulting in different architectures, data pre-processing, and spacings. The second from-scratch baseline, 'No (Fixed)', is an nnU-Net training with the same plans and preprocessing as defined by our pre-training. The dataset-wise mean DSC and mean NSD values across our test dataset suite are provided in Table 3. Additionally, ranking stability of the methods is evaluated through bootstrapping and is provided in Fig. 3.

### 4.1 OBSERVATIONS

**SSL pre-training works** Across all tested datasets, SSL pre-trained methods demonstrate improved downstream segmentation performance. Comparing our S3D-B method to the most similar from-scratch baseline 'No (Fixed)', we observe higher DSC scores in 10 out of 11 test datasets, with an average increase of +2 DSC points and +1.6 NSD points. This improvement is not limited to our method; MG and VF also achieve higher performance than the baseline, indicating the utility of SSL methods when applied to sufficient data and a state-of-the-art architecture.

**MAEs dominate** Throughout our test dataset pool, SSL schemes using the masked image modeling paradigm (MG, S3D-B, and S3D-L) consistently rank higher than the contrastive VoCo or the pseudo-segmentation-based VolumeFusion pre-training method for CNN pre-training. Given the age

| SSL Method | No (Dyn.) | No (Fixed) | VoCo | VF | MG | S3D-B | S3D-L |
|---|---|---|---|---|---|---|---|
| **Dataset** | **Dice Similarity Coefficient (DSC)** | | | | | | |
| MS FLAIR (D1) | 57.81 | 59.82 | 59.70 | 59.29 | 58.64 | **60.35** | 59.85 |
| Brain Mets (D2) | 63.66 | 56.53 | 56.25 | 61.01 | **65.39** | 65.24 | 64.81 |
| Hippocampus (D3) | 89.18 | 89.24 | 88.78 | 89.03 | 89.38 | **89.60** | 89.34 |
| Atlas22 (D4) | 63.28 | 65.52 | 62.97 | 65.76 | 65.93 | **66.95** | 64.58 |
| CrossModa (D5) | **85.64** | 83.44 | 83.07 | 84.24 | 83.91 | 84.08 | 84.02 |
| Cosmos22 (D6) | 60.28 | 78.17 | 77.40 | **80.09** | 79.67 | 80.00 | 80.01 |
| ISLES22 (D7) | 77.94 | 79.44 | 78.14 | 78.96 | 78.85 | 79.70 | **79.89** |
| Hanseg (D8) | 59.00 | 61.85 | 57.47 | 61.49 | **62.52** | 62.11 | 61.93 |
| HNTS-MRG24 (D9) | 66.73 | 65.90 | 67.65 | 63.34 | 68.00 | **68.62** | 67.94 |
| BRATS24 Africa (D10) | **93.07** | 92.51 | 91.97 | 92.16 | 92.36 | 92.19 | 92.90 |
| T2 Aneurysms (D11) | 46.76 | 41.97 | 40.16 | 44.96 | 45.48 | **47.26** | 44.15 |
| Avg. DSC | 69.40 | 70.40 | 69.41 | 70.94 | 71.83 | **72.37** | 71.77 |
| Avg. Rank | 4.64 | 4.55 | 6.27 | 4.36 | 3.18 | **2.00** | 3.00 |
| **Dataset** | **Normalized Surface Distance (NSD)** | | | | | | |
| MS FLAIR (D1) | 78.77 | 80.16 | 79.70 | 79.57 | 79.16 | 80.03 | **80.40** |
| Brain Mets (D2) | 80.72 | 76.72 | 72.77 | 79.20 | 81.51 | **82.53** | 82.32 |
| Hippocampus (D3) | **99.46** | 99.42 | 99.43 | 99.46 | 99.39 | 99.46 | 99.44 |
| Atlas22 (D4) | 70.52 | 73.77 | 70.15 | 73.67 | 74.22 | **75.35** | 73.45 |
| CrossModa (D5) | **99.85** | 99.76 | 99.72 | 99.78 | 99.74 | 99.81 | 99.80 |
| Cosmos22 (D6) | 72.60 | 96.47 | 94.48 | 96.89 | 96.95 | **97.45** | 96.75 |
| ISLES22 (D7) | 88.55 | 90.45 | 89.39 | 90.28 | 89.59 | 90.59 | **90.72** |
| Hanseg (D8) | 82.20 | 85.94 | 80.44 | 85.29 | 85.94 | 85.80 | **86.20** |
| HNTS-MRG24 (D9) | 71.83 | 71.26 | 73.47 | 67.88 | 73.22 | **74.07** | 73.17 |
| BRATS24 Africa (D10) | 95.66 | 95.36 | 94.95 | 94.94 | 95.33 | 95.06 | **95.72** |
| T2 Aneurysms (D11) | **62.24** | 55.56 | 51.79 | 58.97 | 59.38 | 61.18 | 57.07 |
| Avg. NSD | 82.04 | 84.08 | 82.39 | 84.17 | 84.95 | **85.58** | 85.00 |
| Avg. NSD Rank | 4.27 | 4.27 | 5.82 | 4.64 | 4.00 | **2.18** | 2.82 |

Table 3: **S3D out-performs all baselines:** Mean DSC and NSD results across the test datasets, representing a broad selection of brain MR tasks, are presented. 'No Fixed' represents a from-scratch baseline sharing the same architecture, preprocessing and downstream training steps as all SSL methods. 'No Dyn.' represents the original nnU-Net adapted to each downstream dataset individually.

of 'Models Genesis' - published in 2019 - it is surprising to see it outperform the more recent VoCo or VF. We attribute this to a combination of two factors: 1. Models Genesis was originally published and trained on an outdated 3D-UNet (Çiçek et al., 2016) and outside of the powerful nnU-Net framework (Isensee et al., 2021). This highlights the importance of avoiding Pitfall 2: Training on a state-of-the-art backbone. 2. VoCo and VF were introduced in conjunction with architectures they were optimized for. By transferring them to a CNN setting, hyperparameters chosen to optimize the method for their original architecture-pretraining combination may be suboptimal for the new CNN backbone.

S3D-B ranks first with respect to DSC and Normalized Surface Distance (NSD), while S3D-L and MG share the second place. Although S3D-L and MG are very close in Average DSC, S3D-L consistently achieves lower ranks across all datasets. Moreover, according to the bootstrapped aggregated rank (Fig. 3), this superior ranking is reliably higher than that of MG. These results indicate the overall efficacy of our pre-training method compared to currently established methods for CNN pre-training.

**Impact of dynamic configuration** Comparing the 'No (Dyn.)' and 'No (Fixed)' configurations, both trained from scratch, reveals that selecting the appropriate configuration for each dataset can significantly influence performance. For instance, on datasets D2 and D11, the dynamic configuration outperforms the fixed by +7 and +5 DSC points, respectively, while for D6, the fixed configuration yields results +18 DSC points higher. In the majority of datasets where the fixed configuration underperforms relative to the dynamic nnU-Net, pre-training helps to recover performance. However, in some cases, such as with D5, the dynamic default nnU-Net still proves superior.

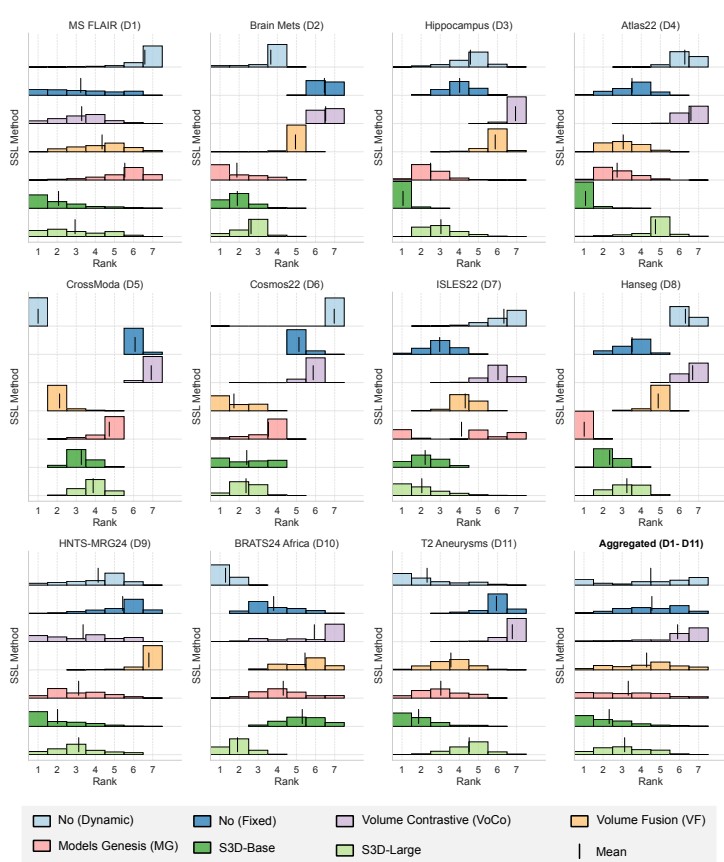

Figure 3: **Dataset rank stability** In addition to absolute mean performance, we report the ranking stability of the methods through bootstrapping for all test datasets as well as the aggregated rank across all datasets.

## 4.2 ABLATION EXPERIMENTS

**Low-Data Regime**   While previous experiments focused on comparing SSL pre-training against a from-scratch baseline with full-scale datasets available, many applications in the medical domain have access to only a very small amount of labeled images. To measure the benefits of pre-training in such a low-data regime, we artificially reduced the total amount of data available for training to 10, 20, 30, or 40 labeled images.

Results are presented in Table 4. It is evident that our pre-trained **S3D-B** model leads to better downstream performance compared to training from scratch in this setting. With just 40 trained images, the fine-tuned model nearly matches the performance of the from-scratch model trained on the full dataset. Future research could explore whether optimizing the training duration or learning rate schedule could prevent the pre-trained network from overfitting during fine-tuning on such a limited number of training images.

**Generalization Performance**   To assess the generalization capability of the proposed pre-training method, we tested two scenarios. First, we evaluated fine-tuning our method on an unseen modality using the TOF Angiography Aneurysms dataset (D12). As shown in Table 7, without pretraining, the fixed configuration suffers a performance drop of 20 Dice points. We attribute this to the significant difference in median spacing for the downstream task ($[0.50, 0.43, 0.43]$ mm), which has a higher resolution than the fixed target spacing of $[1, 1, 1]$ mm used in the pre-training experiments. This lower resolution likely increases the difficulty of segmenting small aneurysms. Despite this decrease, pre-training mitigates some of this degradation and proves highly beneficial compared to training the same configuration from scratch. Interestingly, ModelGenesis achieves the best results, potentially

due to its use of intensity augmentations during pre-training, which increases robustness against brightness shifts, such as when generalizing to different MRI sequences. Second, we fine-tuned on the D2 dataset using only the T1 contrast-enhanced (T1ce) sequence and applied these models directly to D13 without any additional fine-tuning. While the dynamic configuration performed best on the in-distribution validation cases of D2, the results on D13 indicate that MAE pre-training improves generalization across different centers, with our S3D model yielding the best performance.

**Pre-training Time**    Previous studies have demonstrated the positive impact of extended training schedules on the quality of learned representations for downstream tasks (He et al., 2022; Feichtenhofer et al., 2022). To explore the relevance of this factor in the 3D medical domain, we conducted a similar experiment, evaluating training durations ranging from 62.5k to 1M steps. Our results indicate that the benefits of longer training schedules begin to degrade after 250k steps, as evident in Table 5. This could explain why our scaled model did not achieve further performance improvements.

**Fine-tuning length**    Initializing from pre-trained weights has the potential to reduce the computational resources needed for the network to adapt to new tasks. To assess this, we tested different pre-training durations on our development datasets. While maintaining 12.5k iterations for both the decoder and full network warm-up phases, we experimented with varying subsequent training lengths. As shown in Table 8, adding just 12.5k additional iterations (37.5k total) already outperforms training from scratch. However, achieving optimal performance still requires completing the full fine-tuning schedule.

**Multi-Channel Input**    In many medical examinations, it is common to perform multiple scans, as clinicians often require images with different characteristics for accurate decision-making. Consequently, some datasets, like D2, D8, and D10, contain multiple input modalities. While pre-training may involve all modalities, we only feed one modality at a time into the network since not all patients have scans in every modality. This raises the question of how to handle datasets with multiple registered images. To address this, we conducted a 5-fold cross-validation on the D2 development dataset. We evaluated the replication of each input modality along with random initialization of the input stem weights. Additionally, we tested freezing the stem weights during the decoder warm-up phase. As shown in Table 6, the most stable and consistently effective approach was replicating the pre-trained stem and keeping it frozen during the decoder's warm-up period.

## 5    CONCLUSION

This work is the first to demonstrate the potential of properly configured MAEs in 3D medical image segmentation. By overcoming key pitfalls in previous research, such as limited dataset sizes, outdated architectures, and insufficient evaluation, we show a consistent performance improvement over previous SSL methods. Notably, for the first time, we achieve consistent gains over the dynamic, dataset-adaptive nnU-Net baseline, validated across a large and diverse set of development and testing datasets. While our findings are promising, several avenues remain open for future exploration. Notably, increasing training time and batch size did not lead to performance gains, but the question remains of whether scaling the pre-training dataset size or the model parameters could unlock new potential. Furthermore, the intensity shifts employed by ModelGenesis SSL task hint at intriguing possibilities for improving generalization across unseen MRI modalities, which needs to be further explored for MAEs. Lastly, a data-centric approach to curating the most relevant data for SSL represents an exciting frontier for future research. Raw clinical datasets often contain images not intended for diagnostic purposes, such as those used for scanner calibration, which can dilute the effectiveness of pre-training. While we applied basic filtering to exclude low-quality data, more sophisticated filtering techniques could significantly enhance the quality of the pre-training process.

This work follows the spirit of prior studies like nnU-Net Isensee et al. (2021), or XXX, showing that a robust development strategy, informed model configuration, and rigorous validation lead to true and sustainable performance gains, contrasting the current hype for employing and modifying the latest network architecture. With our dynamic open-source framework, we hope to contribute to a cultural shift in the community towards validation-driven development enabling true scientific progress.

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

## A  METHOD CONFIGURATION

Across all baseline methods, we utilize a common set of hyperparameters. For all baseline methods we utilize the same pre-training dataset with the same image preprocessing. Moreover we use the same amount of pre-training steps (250k) as for our S3D-B method and the same fine-tuning scheme, as highlighted in Table 8. Aside from this, we employ the SGD optimizer with LR 1e-2 with a PolyLR schedule, momentum 0.99 and weight decay 3e-5 across all pre-training experiments, as they showed to be highly robust and reliable in the supervised medical image segmentation setting using CNNs (Isensee et al., 2021). Moreover, we denote that all these methods have their backbones replaced with a ResEnc U-Net to minimize confounding effects of different architectures.

### A.1  MODELS GENESIS

Models genesis (Zhou et al., 2021) pre-text task is centered around restoring original patches from transformed versions. The transformed version is achieved by applying four different transformations in various combinations, with the following transformations: a composition of four separate pre-training schemes: (i) **Non-linear intensity transformation:** Alters the intensity distribution while preserving the anatomy, focusing on learning the appearance of organs. (ii) **Out-painting:** Removes part of the image and requires the model to extrapolate from the remaining image, forcing it to learn the global structure of the organs. (iii) **In-painting:** Masks a part of the image, and the model learns to restore the missing parts, focusing on local continuity and context. After having transformed the original image through these augmentations, the model is trained to recover the original image through a convolutional encoder-decoder architecture. This approach consolidates different tasks (appearance, texture, and context learning) into one unified image restoration task, making the model more robust and generalizable.

**Model specific Hyperparameters:**  The entire set of hyperparameters of Models Genesis are contained within the data-augmentation. This allows us to transfer this transformation pipeline, as provided in the official https://github.com/MrGiovanni/ModelsGenesis without any changes to the hyperparameters.

### A.2  VOLUMEFUSION

Volume Fusion (Wang et al., 2023) is a pseudo-segmentation task using two sub-volumes from different 3D scans, which are fused together based on random voxel-level fusion coefficients. The fused image is treated as input, and the model predicts the fusion category of each voxel, mimicking a segmentation task. Pretraining is optimized using a combination of Dice loss and cross-entropy loss.

**Method specific parameters:**  Volume Fusion has unique parameters defining the size ranges of the rectangles used for fusing together images. In our experiments we utilize a rectangle size range between [8, 100] sampled uniformly for each axis. This represent the 62.5% of our input patch size, and identical percentage as in the original paper. Moreover the amount of rectangles sampled is an important parameter. Like in the original paper we sample M $\tilde{U}$(10, 40) different rectangles, iteratively. Lastly, the number of categories was chosen to be 5, as in the original paper (this represent $K = 4$).

### A.3  VOCO

The 'Volume Contrastive Learning Framework' (VoCo) (Wu et al., 2024) is designed to enhance self-supervised learning for 3D medical image analysis by leveraging the consistent contextual positions of anatomical structures. The method involves generating base crops from different regions of 3D images and using these as class assignments. The framework then contrasts random sub-volume crops against these base crops, predicting their contextual positions using a contrastive learning approach. The authors utilize a Swin-UNETR model architecture, employing the AdamW optimizer with a cosine learning rate schedule for 100,000 pre-training steps. The specific hyperparameters include cropping non-overlapping volumes with a size of 64x64x64, and generating 4x4 base crops

| SSL Method | N Train | D1 | D2 | D3 | D4 | D5 | Avg. D1-D5 |
|---|---|---|---|---|---|---|---|
| Scratch | 10 | 40.78 | 43.52 | 84.94 | 44.11 | 76.66 | 58.00 |
| | 20 | 44.46 | 59.46 | 86.75 | 46.33 | 78.67 | 63.13 |
| | 30 | 45.42 | 64.20 | 87.14 | 48.22 | 78.47 | 64.69 |
| | 40 | 49.37* | 60.13 | 87.59 | 50.43 | 78.37 | 65.18 |
| | full | 49.37 | 69.13 | 88.78 | 60.74 | 81.33 | 69.87 |
| S3D-B (ours) | 10 | 43.48 | 48.44 | 84.12 | 41.51 | 77.70 | 59.05 |
| | 20 | 46.58 | 65.30 | 86.61 | 45.50 | 79.52 | 64.70 |
| | 30 | 48.12 | 68.41 | 86.77 | 51.62 | 78.88 | 66.76 |
| | 40 | 51.49* | 72.91 | 87.46 | 53.05 | 80.82 | 69.15 |
| | full | 51.49 | 74.01 | 88.83 | 62.39 | 81.54 | 71.65 |

Table 4: **Forty images with SSL are almost as good as all data from-scratch! S3D-B** model almost reaches the performance of the model trained from-scratch with only 40 training cases. * D1 has only 38 training cases for the train split.

Table 5: **Pre-training length ablation:** Longer pre-training does not lead to improved performance. Interestingly, when exceeding 250k steps.

| PT Iterations | D1 | D2 | D3 | D4 | D5 | Avg. D1-D5 | Train Time [h] |
|---|---|---|---|---|---|---|---|
| 62.5k | 49.49 | 70.79 | 88.82 | 62.95 | 81.27 | 70.67 | 28 |
| 125k | 50.56 | 70.48 | 88.86 | 62.51 | 81.69 | 70.82 | 56 |
| 250k | 51.02 | 74.07 | 88.91 | 62.81 | 81.50 | 71.66 | 112 |
| 500k | 50.93 | 72.71 | 88.88 | 62.17 | 81.86 | 71.31 | 224 |
| 1M | 50.45 | 71.55 | 88.92 | 62.78 | 81.82 | 71.10 | 448 |

during the position prediction task. This represents an input patch size of 384×384×96 which is rescaled and resized to fit exactly 4x4 64x64x64 crops.

Since our chosen patch size 160x160x160 is incompatible with the 64 cube length, we adjusted our patch size for VoCo to 192x192x64. This accommodates a 3x3 grid of 64x64x64. Unfortunately the 4x4 grid led to exceeding the memory limit hence a reduction was necessary. Moreover we increased the target crop size from 4 originally to 5 and increased the batch size from 6 (default in our other experiments) to 12, to fully utilize the 40GB VRAM of an A100 node.

## B  ADDITIONAL RESULTS

Aside from the quantitative data on the development and test dataset, we provide the quantiative data of the ablation experiments here. The following additional results are provided: 1. Results when fine-tuning in a low-data regime are presented in Table 4. 2. Experiment on how to best transfer weights when transferring to a dataset with more than 1 input channel is provided in Table 6 3. Results on how the pre-training effects generalization is provided in Table 7. 4. Experiment results of investigating if one can reduce the fine-tuning steps are presented in Table 8

| Initialization | Decoder Warm-Up | Fold 0 | Fold 1 | Fold 2 | Fold 3 | Fold 4 | Average | STD |
|---|---|---|---|---|---|---|---|---|
| Replication | Frozen | 72.84 | 64.42 | 66.11 | 62.86 | 62.85 | 65.82 | 4.15 |
| Replication | Unfrozen | 72.68 | 63.07 | 65.60 | 66.02 | 61.08 | 65.69 | 4.39 |
| Random | Frozen | 74.38 | 60.89 | 65.10 | 67.55 | 61.31 | **65.85** | 5.51 |
| Random | Unfrozen | 72.20 | 63.16 | 62.25 | 66.71 | 61.47 | 65.16 | 4.42 |

Table 6: Replicating the pre-trained stem weights and freezing them during the decoder warm-up phase yields the most stable and equally best results.

| Experiment | Setting | No Dyn. | No Fixed | VoCo | VF | MG | S3D-B | S3D-L |
|------------|---------|---------|----------|------|-----|-----|-------|-------|
| Modality shift | TOF Angio. Aneurysms(D12) | 42.61 | 22.76 | 22.32 | 31.21 | 34.60 | 28.72 | 26.90 |
| In Distribution | Brain Mets (D2) | 72.81 | 67.93 | 64.34 | 71.69 | 69.05 | 71.56 | 72.53 |
| Patient shift | Brain Mets (D13) | 64.08 | 61.61 | 56.78 | 63.95 | 64.22 | 64.54 | 64.95 |

Table 7: **Pre-training can improve generalization:** We investigate generalization to a new modality time-of-flight (ToF) MRI (top), and the generalization of a resulting method when translating it to a different clinic (bottom).

Table 8: **Fine-tuning length:** When initializing from our pre-trained checkpoint, it is possible to achieve a large fraction of the final performance after less than 15% of the normal training time. Despite this a full training schedule reaches better performance.

| FT Iterations | D1 | D2 | D3 | D4 | D5 | Avg. D1-D5 |
|---------------|-----|-----|-----|-----|-----|------------|
| 25k | 50.85 | 73.99 | 88.51 | 55.49 | 46.00 | 62.97 |
| 37.5k | 51.69 | 74.03 | 88.85 | 60.22 | 81.68 | 71.29 |
| 50k | 51.13 | 73.53 | 88.93 | 60.14 | 81.92 | 71.13 |
| 75k | 51.41 | 72.80 | 89.08 | 63.14 | 81.83 | 71.65 |
| 150k | 50.95 | 71.28 | 88.96 | 62.51 | 81.92 | 71.13 |
| 275k | 53.10 | 71.24 | 89.14 | 63.55 | 82.53 | 71.91 |

