# OpenReview forum: "Revisiting MAE pre-training for 3D medical image segmentation"
_ICLR.cc/2025/Conference — ICLR 2025 Conference Withdrawn Submission_

### Official Review · Reviewer_nA7J · 2024-10-18

**Soundness:** 2
**Presentation:** 1
**Contribution:** 2
**Rating:** 3
**Confidence:** 4

**Summary:**

This paper presents a framework based on self-supervised learning in which a large dataset of 3D brain MRI images is leveraged. The model resulting from this framework was fine-tuned and evaluated on various down-stream tasks, yielding segmentations more accurate than other state-of-the-art models such as nnUNet.

**Strengths:**

- This work identifies and tackles three issues regarding the evaluation of previous methods: small dataset size, inadequate backbones, and insufficient evaluation.
- The pretrained model was evaluated on several datasets with different down-stream segmentation tasks.

**Weaknesses:**

- Very unclear and unorganized manuscript. I believe that it can be improved substantially. I specified many details in "Suggestions", including the following: previous related works were not described, unclear concepts are not introduced, parts of what should be the conclusion (e.g., that MAEs dominate, SSL pretraining works) are in the "Results and discussion" section, there is no section/subsection where the experiments are clearly described and instead they're mixed with "results and discussion". Another example of mixing: right before the conclusion, in only one paragraph (L508-516), we can find an experiment description, the results, and the discussion, all mixed together.
- Limited novelty.
  - Limited methodological novelty. The framework is based on well-established Masked AutoEncoders "with the recent adaptations introduced by Tian et al. (2023); Woo et al. (2023)".
  - Partially limited application novelty since the pretrained models are not publicly available. Although the code is shared, researchers may not have access to large datasets; L53 reads that there seems to be "a public decrease in the community’s willingness to share data" (I don't agree or disagree with this statement, but this may be only regarding brain MRI).
- In many cases, it is unclear if one approach is better than another because no standard deviations are shown. In other words, it cannot be understood whether a method achieving a 71.66 dice coefficient is actually better than another method achieving 71.35.

**Questions:**

## Questions
- L213: "When masking the input image, CNNs are not able to ignore the masked regions in the same manner as transformers can." Can you elaborate on this? (I also suggest doing so in the paper). Why would you want to ignore the masked regions? My understanding is that, the model should learn how to reconstruct those regions.
- L249: "a [3x3x3] convolution is applied to the feature maps at every resolution except the highest resolution to prepare the representations for decoding". What do you mean by "prepare" here? why do they need to be "prepared"?
- Table 1. What does the underlining indicate?
- Table 2. What does it mean "Tr. Stage"? is it "Fine-tuning stage"?


## Suggestions / Other comments
- The title generalizes to "3D Medical Image segmentation" but the experiments are only on brain MRI. I suggest specifying that in the title.
- In the abstract and introduction, the reader is introduced to the concept of "development dataset" (L20), which, to me, it wasn't clear until much later.
- The contributions listed in the introductions were in the form of "we evade Pitfall X by doing Y". I don't think these are contributions. A contribution is something that you provide to the community, e.g., a novel method, an application, an answer to a research question, etc.
- From the beginning of the paper it is advertised that the dataset size is 44k, although this number also includes the images that were discarded. The pretraining dataset size was 39k images, which is still quite large. I suggest saying that the dataset size was 39k and not 44k. Furthermore, the caption of Figure 1 reads "During pretraining, we carefully develop our method with 44k images" which seems to not be true; the dataset size is 39k.
- Figure 1. The "testing" shows a "frozen" icon, but as far as I understood, the models are partially fine-tuned on the "test datasets".
- Figure 1. It is unclear what "underline" means.
- There is no "Previous work" section. Although in the introduction a lot of previous work have been cited, it was mostly to highlight the deficiencies of that previous work and not to explain what were the previous methods about. "Previous work" also gives context to the paper, and it helps introducing the methods that you will later compare. The methods in Table 1 (VoCo, VF, MG) seem to come out of nowhere, and they're "Previous related works". Also, in Section 4.1 "Observations", it is written "SSL schemes using the masked image modeling paradigm (MG, S3D-B, and S3D-L) consistently rank higher than the contrastive VoCo or the pseudo-segmentation-based VolumeFusion pre-training method for CNN pre-training", but the reader has never been told that those previous related works were based on different strategies, which is very important to understand why those methods were chosen.
- To better illustrate the masking, I suggest including a figure where the reader can see how the input of the models looks like.
- Figure 2. The text and numbers are a bit hard to read. I suggest increasing their size.
- Typo in L266: "Results are presented in Table Table 1b"
- Typo in L269: "(S3D-B="
- Typo in L204: "betweem"
- L306: "MAEs are known to benefit from scaling.". I suggested including a citation.
- I suggested having a separate section or subsection where the experiments and experimental settings are clearly defined.
- The first line of the conclusion reads: "This work is the first to demonstrate the potential of properly configured MAEs in 3D medical image segmentation". However, by googleing "masked auto encoder medical image segmentation" many works pop up (e.g., [1,2,3,4]), and since there was no "previous related work" section, it is not clear if this is really "the first to demonstrate the potential of properly configured MAEs in 3D medical image segmentation"

[1]: Self Pre-training with Masked Autoencoders for Medical Image Classification and Segmentation. ISBI 2023.

[2]. Masked Autoencoders for Unsupervised Anomaly Detection in Medical Images. Procedia Computer Science 2023.

[3]. Advancing Volumetric Medical Image Segmentation via Global-Local Masked Autoencoder. Arxiv 2023

[4]. Self-supervised pre-training with contrastive and masked autoencoder methods for dealing with small datasets in deep learning for medical imaging. Sci. Rep 2023.

---

### Official Review · Reviewer_s7sF · 2024-10-25

**Soundness:** 2
**Presentation:** 1
**Contribution:** 1
**Rating:** 3
**Confidence:** 5

**Summary:**

The authors identified three key issues in 3D medical image computing and proposed corresponding solutions. They employed the Masked Auto Encoders (MAE) method for pre-training the model within the existing framework, achieving better performance compared to previous SSL methods.

**Strengths:**

- The authors conducted a substantial number of experiments.

**Weaknesses:**

- The paper resembles a technical report rather than an academic paper, lacking demonstration of its innovation. It addresses the problem by simply combining methods without discussing the essence of the issue. MAE has already proven its competitiveness in previous work, yet the paper merely applies MAE to the backbone without further exploration.
- The writing quality is poor, especially with the confusing use of symbols (e.g., the confusion of D[1\~9], DS[1\~9], and dataset names). The excessive use of items and textbf (too much bold text reduces readability) and quotes (all right quotes, which can be displayed correctly in $\LaTeX$ using \`word') makes the paper difficult to read.
- The paper lacks a symbolic representation and adequate explanation of the task setup and model, instead focusing extensively on dataset selection and hyperparameter choices.
- The figures are confusing. Figure 1 is hard to understand, appearing to mix development and testing in a workflow without showing the model pipeline. Figure 2 is poorly organized, with excessive whitespace and mixed use of pie and bar charts. Figure 3 seems to be generated directly by a program, lacking good organization and sufficient explanation, with a lot of meaningless whitespace.
- The paper lacks visualizations of some results.
- The experimental section only describes performance improvements and changes in tables without further discussion. The results show that the model does not achieve significant performance gains in many experiments (the large model size yields only slight improvements or none at all), suggesting that simply applying MAE does not produce sufficiently good results, and the authors do not propose better methods.
- From Table 1-a, it can be observed that model performance improves based on some sparsification adaptations, raising doubts about whether the results in Table 3 are achieved by stacking tricks rather than the method itself. Table 1-c shows no performance improvement from scaling, and Table 3 even shows performance degradation due to scaling, without explanation, which is disappointing for the method.

**Questions:**

- Should the captions for the tables be placed above the tables instead of below?
- Should the writing issues and figure problems mentioned in the Weakness section be revised?
- Can an explanation be provided for the performance degradation observed with scaling up (Table 3)?
- Does the final model's performance degrade when sparsification adaptations are not used?

---

### Official Review · Reviewer_bU68 · 2024-11-03

**Soundness:** 2
**Presentation:** 3
**Contribution:** 2
**Rating:** 3
**Confidence:** 4

**Summary:**

This paper aims to benchmark various mask autoencoder (MAE) or masked image modeling (MIM) pretraining configurations to determine the optimal one for 3D medical image segmentation using CNN. It collected a large-scale MRI dataset containing around 40k scans for pretraining. The pre-trained model was then applied and evaluated on 13 datasets/tasks.

**Strengths:**

1. Benchmarking SSL pretraining strategy is absolutely important in all fields of AI, including medical vision.
2. This involves a large-scale pretraining dataset, ~40k brain scans.
3. The downstream evaluation sets are also diverse.
4. The presentation is easy to follow, but it certainly can be further improved.

**Weaknesses:**

The reviewer rated soundness as 2 and contribution as 2, given the following reasons:

Soundness:

1. Given the current experiment setups, it is insufficient to conclude the optimal pretraining strategy.
* The patch size of MAE is a critical parameter, while Kaiming's MAE paper [1] did not ablate on that, some other studies ablated on that parameter and found significant performance differences [2-4]. This paper utilized a patch size of 5x5x5 in the bottleneck, equivalent to 32x32x32 in image space. It seems **too large** for the 3D MAE. Both [3] and [4] indicate in the 3D medical image, a high masking ratio and a small patch size are the key ([3] used a patch size of 16x16x16, [4] used 8x8x8).
* Regarding scaling of MAE pretraining, this paper only investigates having 8x batch size, larger learning rate, and 4x training iterations. Those are not keys to evaluating scaling. Scaling more refers to performance gain with an **increase in data size** and an **increase in model parameters**. On high pretraining iterations, the impact of larger batch size and learning rate may not be significant. Extending training iterations may also not help as the MAE training tends to saturate after prolonged training ([1] Fig 7 upper, 800 vs. 1600 epochs are very close, 84.9 vs. 85.1). So what will be really interesting to see is to ablate on 1. **training on 10%, 25%, 50%, 75%, 100% of 40k pretraining datasets**; 2. **varying model's depth to see how performance changes with model size**. In addition, the naming of S3D-L is very **misleading**, as -L always indicates a larger model (with more parameters) in the ML naming convention.

The above two reasons lead to a rating of soundness of 2, as without experiments on those two perspectives, it is hard to conclude the current manuscript presents the optimal strategy.

Contribution:

The reason for a rate of 2 in the contribution is that the current manuscript, entitled 'Revisiting MAE pre-training for 3D medical image segmentation', did not include any comparison with previous studies that utilized MAE pretraining for 3D medical image analysis, notably [3, 5]. Instead, it only involves comparisons with Model Genesis, Volume Fusion, and VoCo.

The contribution of the current study will be much higher if compared to the existing 3D MAE pretraining framework developed for medical images (i.e., [3,5]).

Others:

The quality of Fig. 2 can be improved.

**Overall**, the reviewer recommends rejection because the technical flaws and a lack of comparison with existing 3D MAE frameworks (as presented above) outweigh the benefits brought by large-scale datasets and diverse downstream evaluations.

* [1]: He, Kaiming, et al. "Masked autoencoders are scalable vision learners." Proceedings of the IEEE/CVF conference on computer vision and pattern recognition. 2022.
* [2]: Xie, Zhenda, et al. "Simmim: A simple framework for masked image modeling." Proceedings of the IEEE/CVF conference on computer vision and pattern recognition. 2022.
* [3]: Chen, Zekai, et al. "Masked image modeling advances 3d medical image analysis." Proceedings of the IEEE/CVF Winter Conference on Applications of Computer Vision. 2023.
* [4]: Zhang, Xuzhe, et al. "MAPSeg: Unified Unsupervised Domain Adaptation for Heterogeneous Medical Image Segmentation Based on 3D Masked Autoencoding and Pseudo-Labeling." Proceedings of the IEEE/CVF Conference on Computer Vision and Pattern Recognition. 2024.
* [5]: Tang, Yucheng, et al. "Self-supervised pre-training of swin transformers for 3d medical image analysis." Proceedings of the IEEE/CVF conference on computer vision and pattern recognition. 2022.

**Questions:**

What is the reason for excluding the existing 3D MAE SSL pretraining frameworks for medical images from Table 3?

---

### Official Review · Reviewer_UFXL · 2024-11-04

**Soundness:** 3
**Presentation:** 3
**Contribution:** 3
**Rating:** 6
**Confidence:** 4

**Summary:**

The paper proposes a SSL framework, called nnSSL, for 3D medical image segmentation based on a MAE strategy and thorough evaluation of various design choices. The paper pretrains on a dataset of 44K private MRI and designs a SSL framework using 5 public datasets and uses 7 public datasets for further evaluation and comparison to SOTA methods.

**Strengths:**

The paper is a very timely and relevant contribution to the field of medical image segmentation where the use of self-supervised pretraining is still in its infancy. The paper is clearly written, and proposes a simple, yet effective framework for pretraining for 3D medical segmentation downstream tasks. The analysis of design choices contains valuable insights. The evaluation is thorough, and compares to the most important baseline methods. While not a novel method, getting the sparse convolution MAE to work in 3D is non-trivial, and making the implementation of this public is a sizeable contribution.

**Weaknesses:**

* (W1) The paper does not provide important details on how the weights are transfered for finetuning. Is finetuning performed in the nnUnet framework? Which augmentations are used when finetuning? Are the learning-rate, augmentations etc. fixed for all evaluation datasets? As noted by the authors, selecting an appropriate configuration for each dataset is important. I assume that the configuration is also dynamic for S3D, however the paper does not contain any mention of how this is achieved with pretrained weights.
* (W2) The authors use a patch size of 160^3 which is significantly larger than most previous works, however does not provide any ablations of the effect of this. The proposed performance gains therefore cannot be ruled out to be mainly from using a larger patch size.
* (W3) The paper lacks references to important related work. Specifically, the authors are suggested to include the following two articles in the related works section:
    - SSL with a convolutional MAE for 3D MRI segmentation on a **public** dataset of 44K MRI scans, which similarly revisits various design choices for CNN pretraining, yet with inferior evaluation: [1]
    - Implementation of Spark-like sparse masking for large-scale 3D medical image pretraining: [2]
* (W4) The notes on scaling and the S3D-L variant is misleading since it does not use a model of larger size, yet is scaled in other ways. This meaningfully departs from the established literature, and the authors are encouraged to find another way of communicating the different training setup. Scaling the model and data sizes are important ingredients in compound scaling, yet none of these are performed.
* (W5) The pretraining dataset is private and only limited information on the nature of this dataset is included. For reproducibility purposes, it would be beneficial for the community if the authors would release checkpoints trained on Brains-45K (similar size to the used dataset) from [1].
* (W6) The abstract mentions pretraining is on a dataset of 44K 3D MRI volumes, however the actual pretraining dataset is 39K volumes after filtering out low-quality data. This discrepancy is misleading.

References:

[1] Munk, Asbjørn, et al. "AMAES: Augmented Masked Autoencoder Pretraining on Public Brain MRI Data for 3D-Native Segmentation." _arXiv preprint arXiv:2408.00640_ (2024).

[2] Tang, Fenghe, et al. "Hyspark: Hybrid sparse masking for large scale medical image pre-training." _International Conference on Medical Image Computing and Computer-Assisted Intervention_. Cham: Springer Nature Switzerland, 2024.

**Questions:**

* How is the finetuning implemented? Does the finetuning use nnUnet or a nnUnet like framework?
* Will the authors release pretrained weights and results on public data, such as Brains-45K?
* The authors use a patch size of 160^3, however this is not standard by nnUNet. What is the performance improvement over using 128 or 96 standard in many previous works?

---

### Note · Authors · 2024-11-13

**Comment:**

Given the Scores and the Reviews, we withdraw the paper from ICLR and will revise it.
We want to thank the reviewers for their time and their mostly constructive feedback.

_Despite withdrawing, we believe some points of criticism are disputable and the following should be noted:_

> Claim of first working MAE baseline without showing "Self-supervised pre-training of swin transformers for 3d medical image analysis." and "Masked image modeling advances 3d medical image analysis." are not working

We are certain that we are the first to show convincing results of MAE pre-training but agree that we should provide additional evidence of SwinUNETR and the other Transformer MAE Baseline being sub-par. We originally believed the known deficiencies of SwinUNETR and transformers in 3D medical image segmentation to be sufficient in itself, but will provide evidence in future versions.

> Claiming to train on 44k volumes is misleading, as we filter down to 39k

This will be reworked in the future version.

> Scaling should increase data and parameters

Originally this scaling was conducted to allow adaptation of the architecture on smaller consumer-GPUs as was stated in the manuscript. Depite this we agree that this scaling is suboptimal and that the naming convention is confusing. We will provide a better scaling scheme/paradigm and in the future.

> Partially limited novelty as pre-trained models are not publicly available

We agree that public pre-trained weights would improve the contribution, hence we will provide pre-trained weights in a future version, created on the public 41k volume large ABCD dataset.

> Missing reference to AMAES: Augmented Masked Autoencoder Pretraining on Public Brain MRI Data for 3D-Native Segmentation.

While we would like to use this publicly available dataset, we want to denote that there is no simple way of obtaining it. Many singular data-usage requests need to be conducted, and singular datasets come with specific hurdles associated with use of their data. E.g. PPMI requires users to get papers administratively reviewer,  `If I seek to publish manuscripts using data from PPMI, I agree to follow the guidelines established and written out in the PPMI Publications Policy, including sending manuscripts to the PPMI Data and Publications Committee (DPC) for administrative review.` https://ida.loni.usc.edu/collaboration/access/appLicense.jsp . Same goes for some datasets like OASIS-3, or ADNI: _"If I publish manuscripts using data from ADNI, I agree to the following:
`On the by-line of the manuscript, after the named authors, I will include ADNI as an author
by using the phrase "for the Alzheimer's Disease Neuroimaging Initiative*" with the asterisk
referring to the following statement and list of names"` Which the original paper even violates https://arxiv.org/pdf/2408.00640

Having said this, we want to thank all the reviewers again for their time and effort.
Cheers

**Withdrawal Confirmation:**

I have read and agree with the venue's withdrawal policy on behalf of myself and my co-authors.